# Effect of Gestational Pesticide Exposure on the Child’s Respiratory System: A Narrative Review

**DOI:** 10.3390/ijerph192215418

**Published:** 2022-11-21

**Authors:** María Isabel Ventura-Miranda, Isabel María Fernández-Medina, Eulalia Guillén-Romera, Rocío Ortíz-Amo, María Dolores Ruíz-Fernández

**Affiliations:** 1Department of Nursing Science, Physiotherapy and Medicine, University of Almería, 04120 Almería, Spain; 2Rafael Mendez University Hospital (Lorca), 30800 Murcia, Spain; 3Department of Psychology, University of Almería, 04120 Almería, Spain

**Keywords:** respiratory tract diseases, pregnant women, pesticides, neonate, congenital anomalies

## Abstract

Background: In recent years, concern has arisen worldwide about the potential adverse effects that could result from early-life exposure to pesticides. Asthma, bronchitis, and persistent cough in children have been linked to gestational exposure to pesticides. The respiratory effects of gestational exposure to pesticides are controversial. The aim of this study was to determine the relationship between pesticide exposure in pregnant women and its effect on the respiratory system of their children. Methods: A narrative review was carried out by means of a search in the main databases. Results: Findings of studies confirmed the effects of pesticides on the child’s health. These substances cross the placenta and become transmitters of exposure to the individual at the most sensitive stage of her development. Conclusions: Chronic exposure to pesticides in fetuses is associated with chronic respiratory symptoms and disease.

## 1. Introduction

Nowadays, there is a global concern about the potential adverse effects on human health of exposure to chemicals [1]. Pesticides, chemical substances intended to prevent, destroy, or control pests, are very frequently used in agricultural settings in Spain [2]. They are also used in human medicine, household setting, to treat construction materials, and for vector control [3]. Humans are constantly exposed to pesticide residues present in the environment, water, and food. These residues are absorbed by the body through the skin, digestive and respiratory systems [4].

High exposure to pesticides is known to be harmful to both humans and animals [5]. Animal studies have shown that exposure to pesticides can cause neurochemical changes [6] and alter the development of neuronal cells [7]. In addition, they can also cause endocrine and reproductive system disruptions, induced fetal growth retardation, and structural congenital anomalies [8]. 

Gestation is a crucial period due to the susceptibility of the developing fetus [4]. Drugs such as chemotherapy and oral selenium supplementation have a negative impact on pregnancy and fertility [9,10,11,12]. Pesticides also have the ability to cross the blood–brain barrier. Fetus and child brain is more vulnerable to neurotoxic effects than adults [13]. In addition, exposure to pesticides during pregnancy could adversely affect fetal development because pesticides can cross the placental barrier and have even been found in amniotic fluid [14]. Other studies have concluded that prenatal exposure to pesticides may be associated with some birth outcomes, such as mental and psychomotor delay and attention problems in school children [15].

The negative effects of pesticides on the respiratory system are well documented. Pesticide exposure has been associated with a reduction in lung inspiratory vital capacity, which may contribute to the development of chronic obstructive pulmonary disease (COPD) and wheezing symptoms [16,17]. Maternal exposure to pesticides has been associated with asthma, bronchitis, and persistent cough in children [18,19]. However, other studies have found no significant differences [20,21]. The evidence has suggested that exposure to pesticides increases the risk of birth defects in the respiratory system, such as orofacial clefts [22]. 

Although there is evidence of the negative impact of prenatal pesticide exposure on the health of the child, the information is inconsistent and limited. Therefore, the aim of this study was to determine the relationship between pesticide exposure in pregnant women and its effect on the respiratory system of their children.

## 2. Materials and Methods

### 2.1. Type of Research

The study carried out was a narrative literature review of articles related to the main aim of the study published during the last fifteen years. 

### 2.2. Study Selection Criteria

In total, 13 studies were selected from a total of 1951 that were found in the search in the different databases.

This work includes bibliographic reviews, cross-sectional studies, cohort studies, and systematic reviews.

#### Inclusion Criteria

We included articles that meet the following requirements: studies in Spanish and English published in the last fifteen years that discuss how pesticides affect pregnant women and their children. Older articles dealing with the subject of the study were included because of their importance.

### 2.3. Procedure for Collecting Information: Search Strategy and Selection of Studies

An electronic search was carried out of different articles and studies in English and Spanish, published in the last fifteen years in the following databases: Pubmed, Lilacs, Dialnet, and Elsevier with the keywords: pesticides, respiratory tract diseases, congenital anomalies, pregnant women, neonates, or children. The Boolean operators AND and OR were used to make the search more comprehensive and specific to avoid unwanted literature. The literature search was conducted independently by two authors.

The article selection process was carried out in three phases. In the first phase, articles were selected based on their title and abstract to subsequently check the type of study design and examine compliance with the selection criteria previously described. They were considered potentially relevant to the topic when the conclusions and results contained information on the effects of pesticides prenatal exposition on the respiratory tract of children. The full-text papers were then reviewed, and the methodological quality of the studies was assessed. The most significant data from all studies were collected and grouped by category using a manual data extraction sheet. The analysis variables extracted were: (i) author/s and year of publication, (ii) aim, (iii) methodology, and (iv) most relevant results.

## 3. Results

The initial search yielded a total of 1951 papers. After eliminating duplicates and screening by title and abstract, 53 publications were reviewed in full text. In total, 13 articles were included in this review (Figure 1). 

### 3.1. Effects of Pesticides on the Respiratory Tract

A cross-sectional study published in 2015 on the effects of pesticides on the respiratory tract at the pesticide-exposed population level found evidence that exposure to arsenic in drinking water during early childhood or in utero is associated with increased respiratory symptoms or disease in adulthood (OR: 1.160; 95%CI: 1.02–1.30; *p*-value: 0.002). This arsenic exposure was associated with impaired lung function and suggested that an adverse effect could be due to a chronic inflammatory response to arsenic. In order to better assess the risk of arsenic exposure, only children in whom an acceptable sputum sample was obtained were included in the study (n = 275), although initially, more than 500 children were evaluated [23]. 

Otherwise, Hauptman et al. (2015) is a systematic review analyzing the epidemiology and evidence of known or proposed mechanisms of environmental, chemical, infectious, and perinatal exposures that relate to the development of pediatric allergy and asthma. The new evidence explores the effects of high- and low-molecular-weight phthalates, such as pesticides, dichlorophenols, and broad-spectrum antimicrobials, of their impact on the development of asthma [24]. 

In research by Raanan et al. (2015), the findings suggest that exposure to organophosphate pesticides since gestation is associated with respiratory symptoms consistent with a possible diagnosis of asthma in a population of children. The effect size estimates were not detailed in the original publication [25].

On the one hand, Sunyer et al. (2005) analyzed any association between prenatal dichlorodiphenyl-dichloroethylene (DDE) and other organochlorine compounds and atopy and asthma during childhood. The results suggest that prenatal exposure to DEE residues may be associated with asthma: RR: 1.46 (95%CI): 0.92–2.32; *p*-value: 0.10. At 4 years of age, wheezing increased with DEE concentration [26]. Another study conducted by the same author showed that DDE at birth may be associated with diagnosed asthma (OR: 1.18; 95%CI: 1.01–1.39) and persistent wheezing (OR:1.13; 95%CI: 0.98–1.30) but not with DEE at 4 years. DDE does not appear to modify the protective effect of breastfeeding on asthma (OR:0.33; 95%CI: 0.08–0.87) and wheezing (OR: 0.53; 95%CI:0.34–0.82). In this study, all women presenting for antenatal care in Menorca, Spain, over a 12-month period beginning in mid-1997, were invited to take part in a longitudinal study that included a yearly visit [27]. 

Mamane et al. (2015) observed that the concentration of DDE blood levels during pregnancy or in the cord blood at birth increases the risk of childhood asthma (OR: 1.18; 95%CI: 1.01–1.39; *p*-value: 0.05) and wheezing (relative risk 2.36; 95%CI: 1.19–4.69; *p*-value: 0.05). However, no significant association was found between pesticide levels in infant blood or breast milk and asthma or respiratory infections in children. In total, 20 studies dealing with respiratory health and non-occupational pesticide exposure were identified, 14 carried out on children and 6 on adults [28]. Their results have been confirmed by the study of Gascon et al. (2014), which concluded that prenatal exposure to DEE is associated with bronchitis, bronchiolitis, wheezing, pneumonia, and chest infection in children between the ages of 6 months and 4 years. Each cohort included births from 1997 to 2008. The HUMIS cohort was restricted to mothers who breastfeed their child for at least 1 month. In total, 4608 live newborns with information on DDE and polychlorinated biphenyl 153 (PCB 153) exposure and at least 1 respiratory health outcome, as defined below, were included [29]. 

Finally, the study by Gilden et al. (2020) found an association between gestational exposure to organophosphate pesticides (OPP) and pyrethroid metabolites (3PBA) metabolites with child respiratory symptoms in participants with genetic susceptibility and lower fruit and vegetable consumption (OR: 1.35; 95%CI: 1.013–1.815, *p*-value: 0.038). This study conducted a secondary data analysis within the Health Outcomes and Measures Of the Environment (HOME) Study, a prospective pregnancy and birth cohort that followed mothers and their children in the greater Cincinnati, Ohio, USA metropolitan area from the second trimester of pregnancy [30].

### 3.2. Association between Pesticide Exposure and Risk of Congenital Airway Abnormalities

Romitti et al. (2007) suggest that maternal exposure to pesticides is associated with a modest but marginally significant risk of cleft palate. The full text of 230 studies was reviewed in detail, and of these, 19 studies were included in the final analysis. Maternal occupational exposure to pesticides was associated with an increased risk of clefting (OR: 1.37; 95%CI: 1.04–1.81) [31]. Yang et al. (2014) were the first to estimate cleft palate risk based on assessment of individual pesticide exposure because of residential proximity to agricultural use. This analysis included study subjects with estimated dates of delivery from October 1997 to December 2006. Another study analyzed early gestational exposures to pesticides and the risk of anencephaly, spina bifida, and clef lip with or without cleft palate (CL/P). Adjusted odds ratios ranged from 1.6 to 5.1. Therefore, it is not possible to establish an association between agricultural pesticide exposures and risks of selected birth defects [32]. Hao et al. (2015) linked different maternal antecedents, such as fever and common cold without fever, paternal smoking, and alcohol consumption together with maternal exposure to organic solvents, heavy metals or pesticides, and multivitamin use during the preconception period and were associated with CL/P and cleft palate only (CPO). Maternal exposure to organic solvents, heavy metals, or pesticides was associated with CL/P (CL/P: 2.15 and 5.04, 95%CI: 1.37–3.38 and 3.00–8.46) and CPO (CPO: 10.65, 7.28 and 3.48; 95%CI: 2.54–44.67, 1.41–37.63 and 1.06–11.46) [33]. Suhl et al. (2018) observed that there were associations mostly close to unity between maternal occupational pesticide exposure and orofacial clefts. Associations for paternal exposure to occupational pesticides were mostly close to or below unity for cleft lip ± cleft palate and mostly positive for cleft palate. The effect size estimate in this study was due to funding that enabled the assessment of parental occupational pesticide exposure for cases and controls with DDD from October 1997 to December 2002 [34].

A study conducted by Lewis et al. (2014) found evidence that demographic factors, green leafy vegetable consumption, and pesticide use are potentially important determinants of exposure to certain pesticides among this group of pregnant women. Significant associations between urinary concentrations of biomarkers and consumption of fruits, vegetables, and legumes were not found. However, women who consumed collards (OR:5.9, 95%CI: 1.3, 26.7) or spinach (OR: 4.4; 95%CI: 1.1, 17.9) had more urinary concentrations of 2,4-dichlorophenoxyacetic acid 2,4-D. On the other hand, participants’ use of pesticides at home increases the urinary concentration of 2,2-dimethylcyclopropane carboxylic acid (*trans*-DCCA) (OR: 4.9; 95%CI:1.1, 22.1). Pesticide use has been associated with a significant increase in respiratory tract symptoms such as asthma, chronic cough, and bronchitis, an obstructive pulmonary disease, especially in agricultural workers, who are exposed to many of these compounds. This analysis concerned 54 pregnant women participating in the Puerto Rico Test site for the Exploring ContaminationThreats (PROTECT) project. PROTECT is an ongoing prospective birth cohort in the northern karst region of Puerto Rico designed to assess the potential relationship between environmental toxicant exposures and the risk of pre-term birth and other adverse pregnancy outcomes [35]. Table 1 shows the main results of the articles included in the review.

## 4. Discussion

The main aim of this literature review was to determine the relationship between pesticide exposure in pregnant women and its effect on the respiratory system of their children. The included articles provide evidence of a directly proportional relationship between prenatal pesticide exposure to respiratory effects in children. 

Among the 13 studies reviewed that studied pregnant women, three of them associated in utero pesticides exposures with an oropharyngeal cleft in children at birth [31,33,34], and another study identified DEET concentrations in the urine of pregnant women who had been exposed to pesticides [35]. Although pesticides may play a teratogenic role, in contrast, other studies have found no significant evidence linking prenatal exposure to birth defects [36]. This could be because not all studies have assessed the influence of the same pesticides on the production of birth defects. In addition, it is likely that not only pesticide exposure but other environmental and genetic factors are involved in the production of oropharyngeal cleft in children at birth.

Pesticide exposure has often been associated with asthma in young children [24,25,26] and wheezing [28]. Survey et al. (2019) [26,27] showed that there was a direct association between prenatal exposure to DDE and other organochlorine compounds and atopy, asthma, and persistent wheezing. Additionally, the protective effect of breastfeeding on asthma may not be modified by exposure to DEE. In the same line, Gascon et al. (2014) [29] concluded that prenatal exposure to DDE increases the risk of respiratory health symptoms in children younger than 18 months. Furthermore, exposure to other components, such as mancozeb and ethylenethiourea, has also shown a positive relationship with children’s respiratory outcomes in the first year of life [37]. However, some authors [30] did not find strong associations between gestational pesticide exposure and specific birth defect phenotypes.

The studies reviewed in many of these articles provide evidence that chemicals placed in the environment by human activity can promote disease by altering gene expression [30,38]. Environmental exposure to pesticides and persistent organic pollutants can modulate the expression levels of different genes (NRXN and PAX6 genes) and induces epigenetic alterations in the modulation of DNA methylation status and in the expression levels of miRNAs [39]. Further studies are needed to analyze the epigenetic alterations induced by pesticide exposure and the mechanisms responsible for the alterations in children with prenatal exposure to pesticides. In the study by Olivas-Calderón et al. (2015) [23], they observed that arsenic exposure negatively affects early stages and significantly increases the frequency of an abnormal spirometric pattern in children. Arsenic is a non-mutagenic human carcinogen, which also induces DNA hypomethylation that is associated with chromosomal instability [40]. Hypomethylation has been linked to cellular oxidative stress, which may be responsible for pesticide-induced health effects [41]. Furthermore, Marsi et al. (2006) showed alterations in miRNA due to arsenic exposure [42]. These arsenic-induced changes could explain the alterations in the respiratory system of children.

It was also shown that exposure to chemicals in early life can influence the development of the immune system and thus affect the health of the child [25,26]. Organochlorine contaminants modulate key genes of the cellular innate immune responses involved in host cell defense against viral infections [43].

Survey et al. (2019) [26,27] showed that there was a direct association between occupational exposure to pesticides and obstructive lung disease. There was also a positive and significant association between pesticide exposure and the risk of chronic bronchitis. There was also a significant association between occupational pesticide exposure and an increased risk of COPD.

To the knowledge of many of the authors of these studies, there is little epidemiology and studies directly related to the effects of pesticide exposure and respiratory effects in children. Since human exposure to pesticides may be associated with many of the effects studied, additional research will increase understanding of the causes of pesticide exposure in pregnant women. All the articles reviewed have the same limitation; there is a need to improve the assessment and increase the sample size to facilitate risk estimation. Furthermore, the studies used different measurement techniques, probably because not all the pesticides measured were sensitive to a single technique. This use of different measurement techniques could limit the results of the studies analyzed.

Many of the studies analyzed relate the involvement of contaminants with dietary susceptibility and maternal genetics [22,33,35].

To better understand the relationship between pesticide exposure and orofacial clefts, future studies should consider the evaluation of multiple parental exposure pathways, etiologically homogeneous phenotypes, and individual genetic susceptibility.

There are not many studies on respiratory problems due to pesticides since most of the articles are related to low birth weight and urogenital alterations, but very little on disorders of the respiratory system. More research is needed to address the remaining uncertainties in this field.

Health professionals should counsel pregnant women, workers, and the population about risk factors, while research continues to find improvements and provide a clearer picture of the magnitude of harmful exposures.

Health professionals are obliged to be aware of the problem, recognize exposures, and warn the population, especially the most vulnerable, about situations of increased risk. In addition, they must be involved in the decision-making processes by participating in the appropriate forums and demanding the most appropriate application of the protection systems at any given time.

As future research lines, more research is needed to understand which aspects of the identified compounds lead to increased exposure and whether exposure during pregnancy is associated with adverse health problems.

## 5. Conclusions

A directly proportional relationship between pesticide exposure to respiratory tract effects has been found in children. Maternal exposure to pesticides is associated with a modest but marginally significant risk of clefting. The embryo/fetus and neonate are extremely sensitive to exposure, and adverse effects are more severe than in adults. Chronic pesticide exposure in children is associated with chronic respiratory symptoms and disease. It is necessary to develop policies aimed at reducing exposure to pesticides of different types that may contribute to the prevention of respiratory diseases among workers and the creation of mechanisms to minimize the exposure of the population.

## Figures and Tables

**Figure 1 ijerph-19-15418-f001:**
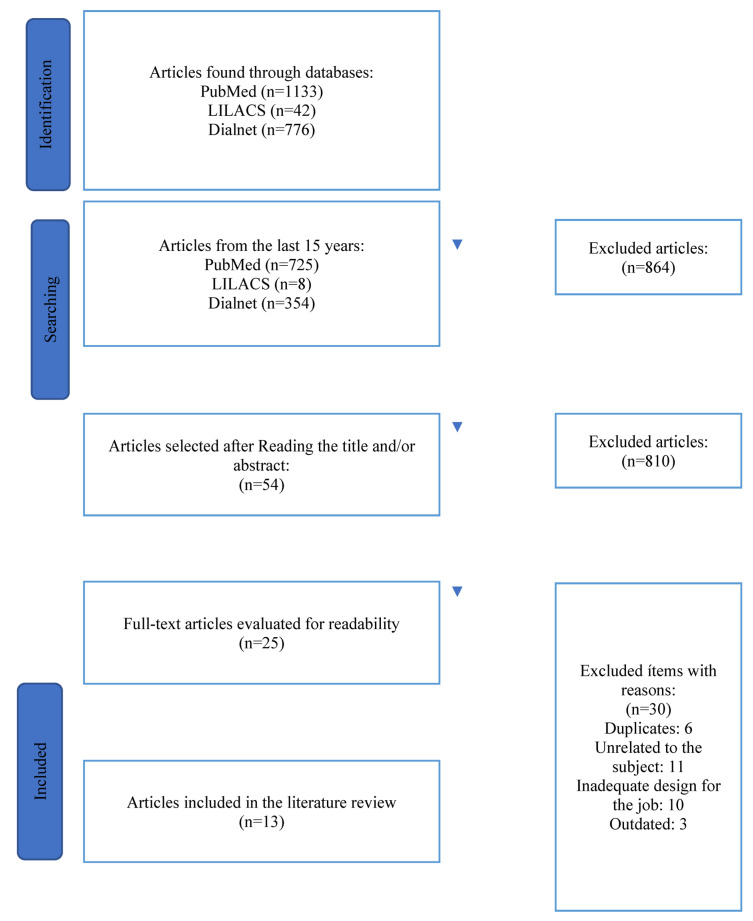
Articles included in the review.

**Table 1 ijerph-19-15418-t001:** Results of the articles included in the review.

Author/Year	Objectives	Methodology	Sample Size	Rural Area/ City/Country	Tissue Sample Source	Evaluated Exposure (Active Matter)	Exposure Measurement Method	Results
Sunyer et al. (2005) [26]	To identify any association between prenatal DDE and other organochlorine compounds and atopy and asthma during infancy.	Longitudinal study	468 children	Menorca is one of the Balearic Islands in the northwest Mediterranean Sea.	In total, 405 (84%) had organochlorine compounds in cord serum measured. Blood was drawn at 4 years of age in 360 children, 306 of whom had IgEs and peripheral white blood cells measured. Asthma was defined based on wheezing at 4 years of age, persistent wheezing, or doctor-diagnosed asthma.	Specific IgE against house dust mite (Der p1), cat (Fel d1), and grass was measured using the CAP method, with levels > 0.34 kU/L being considered positive.	Prenatal DEE and other organochlorines were measured in cord serum by GC with electron capture detection and GC coupled to chemical ionization negative-ion mass spectrometry.	Prenatal exposure to DDE may contribute to development of asthma.
Sunyer et al. (2006) [27]	To investigate the association of DEE with childhood asthma measured up to age 6 and the effect of DEE on the protective effect of breastfeeding on asthma.	Longitudinal study	462 children	Menorca is one of the Balearic Islands in the northwest Mediterranean Sea.	Organochlorine compounds were measured in cord serum of 402 (84%) children. Blood was drawn in 360 children at 4 years of age, 285 of whom had organochlorine compounds measured.	A weal of 2 mm or greater in the presence of a positive histamine control and a negative uncoated control constituted a positive skin test. A positive skin test to at least one allergen (Der p 1, Der f 1, cat, dog, grass pollen, mixed tree, mixed graminae, parietaria) was considered indicative of atopy.	DDE and DDT were measured in cord serum by GC with electron capture detection and GC coupled to chemical ionization negative-ion mass spectrometry	DDE does not modify the protective effect of breastfeeding on asthma. Asthma and persistent wheezing were associated with DDE at birth but not with DDE at 4 years.
Romitti et al. (2007) [31]	The risk of orofacial clefts associated with pesticide exposure was examined by performing a meta-analysis of studies published between 1966 and 2005.	Systematic review	19 studies					Many of the studies identified as suitable for analysis used a retrospective design with different sample sizes, levels of exposure assessment, and phenotypic assessment. Maternal exposure is associated with an increased risk of clefting.
Lewis et al. (2014) [35]	To describe distributions, evaluate subjects within temporal variability, and identify predictors of urinary DEET concentrations in pregnant women in Puerto Rico.	Cohort study	54 pregnant women	Puerto Rico	Urinary concentrations	At three separate time points (20 ± 2 weeks, 24 ± 2 weeks, and 28 ± 2 weeks of gestation).	Measured urinary concentrations of the insect-repellent DEET and two of its metabolites, four pyrethroid insecticide metabolites, and two chlorophenoxy herbicides.	These distributions were calculated and compared with those of women of reproductive age in the US population. Pesticide use in a group of pregnant women was shown to be associated with exposure to several pesticides
Yang et al. (2014) [32]	Use population-based data on specific birth defects.	Cases and controls	73 cases with anencephaly, 123 with spina bifida, 277 with CLP, and 117 with cleft palate only, in addition to 785 controls.	San Joaquin Valley, California.	Exposure estimates were based on residential proximity to agricultural pesticide applications during early pregnancy.	Chemical groups included petroleum derivatives for anencephaly, hydroxybenzonitrile herbicides for spina bifida, and 2,6-dinitroaniline herbicides and dithiocarbamates-methyl isothiocyanate for CLP. The specific chemicals included 2,4-D dimethylamine salt, methomyl, imidacloprid, and α-(para-nonylphenyl)-ω-hydroxypoly(oxyethylene) phosphate ester for anencephaly; the herbicide bromoxynil octanoate for spina bifida; and trifluralin and maneb for CLP.	A total of 38% of the subjects were exposed to 52 chemical groups and 257 specific chemicals.	Results have been inconsistent. Some associations have been observed between gestational pesticide exposure and specific birth defect phenotypes, but the data are insufficient to draw clear results.
Mamane et al. (2015) [28]	To test the respiratory effects of environmental exposure to pesticides on children and adults.	Literature review	20 studies					Studies have identified that prenatal exposition to pesticides increases the risk of asthma and wheezing in young children.
Gascon et al. (2014) [29]	To examine the association between prenatal exposure to DDE and 153 PCB and children’s respiratory health in European birth cohorts.	Cohort study	4608 mothers and children	7 European countries.		Studied the association between prenatal exposure to DDE and PCB 153 and children’s respiratory health in European birth cohorts.	Modeling occurrences of the outcomes on the estimates of cord-serum concentrations of PCB 153 and DDE as continuous variables (per doubling exposure) and as cohort-specific tertiles.	The prenatal DDE exposure increases the risk of respiratory health symptoms in children below 18 months.
Olivas-Calderón et al. (2015) [23]	To demonstrate that arsenic exposure during early childhood or in utero in children was associated with impaired lung function.	Cross-sectional study	275 healthy children.	Located in the north-central part of Mexico.	Cross-sectional study in a cohort of children associating lung inflammatory biomarkers and lung function with urinary.	Arsenic urinary levels.	Inflammation biomarkers were measured in sputum by ELISA, and the lung function was evaluated by spirometry.	Arsenic exposure negatively affects during the early stages and significantly increases the frequency of an abnormal spirometric pattern in children. Fifty-eight percent of the children studied were found to have a restrictive spirometry pattern.
Hauptman et al. (2015) [24]	To discuss the epidemiology and evidence of known or proposed mechanisms of environmental, chemical, infectious, perinatal, and infectious exposures that relate to the development of pediatric allergy and asthma	Systematic review						The new evidence explores the effects of high- and low-molecular-weight phthalates, such as pesticides, dichlorophenols, and broad-spectrum antimicrobials, and their impact on the development of asthma.
Raanan et al. (2015) [25]	To investigate the relationship between childhood exposure to organochlorine pesticides and respiratory outcomes.	Cohort study	359 mothers and children.	Chamascos, Mexico.	Urine.	DAP metabolites of OP pesticides, specifically DE and DM metabolites.	Twice during pregnancy (mean = 13 and 26 weeks gestation) and from children five times during childhood (0.5–5 years).	Prenatal concentrations of organophosphate pesticides present respiratory symptoms compatible with asthma in infancy.
Hao et al. (2015) [33]	To test the possible association of possible parental environmental exposures and maternal supplement intake with the risk of nonsyndromic orofacial clefting	Retrospective study	499 cases and 480 controls.	Heilongjiang Province.	Extracted information on case and control mothers from interviewer-administered questionnaires. The interview was administered by a trained interviewer and addressed exposures from one month before conception through the end of the first trimester.	Women were asked whether they used multivitamins, folic acid supplements, or cod liver oil during the one-month preconceptional period or first trimester.	Mothers were defined as being exposed to organic solvents when reporting contact with industrial cleaning products (degreasers), paints, printing inks, or glues in their jobs. They were considered exposed to heavy metals (cadmium, cobalt, or lead) exposure if their jobs involved production of pigments or batteries, galvanization, or recycling of electric tools. Mothers were considered exposed to pesticides if they were engaged in agriculture during the periconceptional period.	The results showed that maternal history of fever and common cold without fever, paternal smoking and alcohol consumption, maternal exposure to organic solvents, heavy metals or pesticides, and multivitamin use during the preconception period were associated with cleft lip or no cleft palate and cleft palate only.
Gilden et al. (2020) [30]	To examine the association of gestational urinary OP and 3PBA concentrations with child wheeze, forced expiratory volume in one second at ages 4 and 5 years, and wheeze trajectory patterns through age 8 years.	Prospective pregnancy and birth cohort study	Total of 468 pregnant women were enrolled between March 2003 and January 2006, and 390 mothers delivered live-born singletons who were followed from birth to age 8 years.	The greater Cincinnati, Ohio, USA metropolitan area.	Mothers provided urine samples twice during pregnancy, at 16 and 26 weeks gestation.	Inclusion criteria during enrollment of pregnant women included: 16 ± 3 weeks gestation, ≥18 years old, living in a home built before 1978 (this was to focus the cohort on potential lead exposure), no history of HIV infection, and not taking medications for seizure or thyroid disorders.	The HOME Study was designed to assess the relationship between low-level environmental chemical exposures and many aspects of child health, including child respiratory outcomes.	Gestational OP and 3PBA metabolites were linked to childhood respiratory symptoms in participants with maternal and genetic susceptibility.
Suhl et al. (2018) [34]	To examine the relationship between parental occupational pesticide exposures and nonsyndromic orofacial clefts in their offspring.	Cases and controls	Examined risk factors for over 30 major structural birth defects among deliveries from October 1997 through December 2011. Approximately 100 controls per year per site were recruited.	Arkansas, California, Georgia, Iowa, Massachusetts, North Carolina, New Jersey, New York, Texas, and Utah.	Live births were followed for 1 year to confirm diagnosis.	Nonsyndromic OFCs.	Parental occupational exposures to insecticides, herbicides, and fungicides, alone or in combinations, during maternal (1 month before through 3 months after conception) and paternal (3 months before through 3 months after conception) critical exposure periods between orofacial cleft cases and unaffected controls.	This study observed associations mostly close to unity between maternal occupational pesticide exposure and orofacial clefts. Associations for paternal occupational pesticide exposure were mostly close to or below unity for cleft lip ± cleft palate and mostly positive for cleft palate.

DEET: N-N-diethyl-meta-toluamide; DDE: dichlorodiphenyl-dichloroethylene; DDT: dichlorodiphenyltrichloroethane; GC: gas chromatography; PCB 153: polychlorinated biphenyl 153; OP: organophosphate; 3PBA: pyrethroid metabolite; CLP: cleft lift palate; HIV: human immunodeficiency virus; OFC: orofacial clefts; DAP: Dialkyl phosphate; DE: diethyl phosphate; DM: dimethyl phosphate.

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
