# Peer review of "Effect of Gestational Pesticide Exposure on the Child’s Respiratory System: A Narrative Review"

_ijerph, 2022, doi:10.3390/ijerph192215418_

Round 1
Reviewer 1 Report (New Reviewer)
Dear Authors,
Please, take these suggestions into consideration:
1- ABSTRACT.
Please, divide the abstract into correct sections for a research article (i.e. background, methods, results, ..). Also, please include the main examples of chronic respiratory symptoms and disease secondary to gestational pesticides exposure.
2- INTRODUCTION. Line 44 “The negative effects of pesticide on the respiratory system are well documented.”
Please, add examples of the negative effects of pesticide exposure in the adults’ lung as well, with some references.
3- STUDY SELECTION CRITERIA. Line 58. “Thirty-two studies were selected...”
Please, correct this sentence, since it contradicts the paragraph “Results”, in which it is stated that the articles included in the review are 14.
4- FIGURE 1. Please, substitute the word “antiquity” with “outdated” or “obsolete”.
5- RESULTS. Effects of pesticides on the respiratory tract. Lines 129-131 “in research by Raanan et al. (2015)...children”.
Please, revise this sentence: in the cited article, exposure to organophosphate pesticides occurred since gestation, not only during childhood. It is important to highlight this information in order to avoid misinterpretations, since it is strictly linked to the aim of this study.
6- RESULTS. Effects of pesticides on the respiratory tract. Lines 132-137 “On the one hand, Doust et al. (2014)...including chronic bronchitis”.
The review of Doust et al. (2014) included both studies on children and adults exposure to pesticides and pregnant women exposure. Since the aim of this study is to analyze the effect of pesticide exposure to pregnant women and eventual effects on their children’s respiratory system, I suggest to focus on the studies included in the review of Doust et al. which focus on this topic (Sunyer J, Torrent M, Munoz-Ortiz L, et al. Prenatal dichlorodiphenyldichloroethylene (DDE) and asthma in children. Environ Health Perspect 2005; 113: 1787–1790; Sunyer J, Torrent M, Garcia-Esteban R, et al. Early exposure to dichlorodiphenyldichloroethylene, breastfeeding and asthma at age six. Clin Exp Allergy 2006; 36: 1236–1241.)
7- RESULTS. Effects of pesticides on the respiratory tract. Lines 137-140 “Pourhassan et al. (2019)...rhinitis.”
Both studies (Pourhassan et al. 2019 and Slager et al. 2009) do not meet the aim of this review, since they do not address the research topic, so they should not be included.
8- DISCUSSION. Lines 186-187. “Slager et al...rhinitis”.
Please, remove this sentence, according to point 7.
9- DISCUSSION. Lines 197-201. “Pourhassan et al...COPD”.
Please, remove this sentence, according to point 7.
Author Response
Thank you for your revision.

Reviewer 2 Report (New Reviewer)
Major.
Methods. The literature search was conducted independently by several authors to check possible bias?
Methods. The authors included literature reviews. To what extent the findings from previous literature reviews overlap with the other study types considered by the authors?
Results. For clinical utility, specify effect size estimates, confidence interval, and p-values for the reported associations, when available. Specify when these are not available to contextualize a limitation of the studies conducted to date. Moreover, additional columns detailing sample size, rural area/city/country, tissue sample source, evaluated exposure (active matter), and exposure measurement method should be included in Table 1 for comprehensibility.
Results. The authors summarize the main findings of the articles in terms of the association founds, but they do not discuss any potential biological mechanism affected by pesticide exposure. This could discuss the possible implications of these contaminants at different biological layers (genetics, epigenetic, transcriptomics,...) in the results or discussion.
Discussion. May these studies be biased because of measurement techniques?
Discussion. May the authors comments about routes of exposures (inhalative route, dermal or oral route), as well as other factors influencing exposure (i.e., repeated exposure, substance kinetics?)
Minor.
Abstract. Consider being more precise by detailing the “toxic effects of pesticides on the child’s health”.
Methods. Please revise that the syntaxis of the paragraph at epigraph 2.2 is correct.
Figure 1. The figure has no footnotes. Moreover, could the authors clarify their definition on antiquity (Figure 1).
Discussion. Could the authors expand of advantages and disadvantages of these studies and what to consider in future research in the field?
Author Response
Thank you for your revision.

Round 2
Reviewer 1 Report (New Reviewer)
Dear Authors,
I am satisfied with the improvements you made with your paper, according to my suggestions.
I just want to point out that Figure 1 and Table 2 need formatting.
Author Response
Thank you for your revision. The changes have been made.
Reviewer 2 Report (New Reviewer)
The authors are encouraged to think thoroughly about the optimal manner to address the reviewer’s comments by providing statements that are supported by arguments. On the other hand, I congratulate the authors for the efforts on improving Table 1.
Major comment.
1. Overall, this manuscript summarizes the findings of the literature but fails to discuss what these studies were lacking in depth, what could be improved in future studies, or what should be the focus of the future studies in the field in a concise manner. The discussion section can be improved to be more precise. For example: “three of them associated in utero exposures [to what?] with oropharyngeal cleft in children at birth”. Moreover, the authors state “that chemicals placed in the environment by human activity can promote disease by altering gene expression”, but they did not mention what genes were affected and what these are involved in the underlying biological pathways in the example they cite later (Olivas-Calderón et al. (2015)”. This is a missed opportunity to summarize briefly the effects that the relevant pesticides may be known to have in different omic layers (transcriptomics, proteomics, epigenetics), or to support the need of more studies in case these studies have not been performed before. In addition, there are several one sentence paragraph in the discussion. Al this together seems to point out that the discussion is a mere list but not a full discussion of the findings of the search in the context of the field.
2. Some parts of the review are copy pasted from previous publications: eg. “More than 500 children were evaluated; however since satisfactory sputum sample is not easy to obtain at this group-age, we only include in this study those children where acceptable sputum samples were obtained (n=275) allowing us to better assess the risk to children from arsenic exposure.”
3. State the databases that were consulted to select the articles (eg. Web of science, Pubmed) in the methods section.
4. I’m not sure what the authors mean with “CI=95% and the p-value < 0.05”. They should specify the values in the text and to what these refers to. For example, Pesticides exposure in children with asthma was associated with higher risk of XXX: OR (95%CI): 1.45 (1.16-1.50); p-value: 1x10-4). This type of data provides a measurement of clinical utility for the readers.
5. Please revise that references are cited when needed throughout the text. For example, “Many of the studies analyzed relate the involvement of contaminants with dietary susceptibility and maternal genetics”, but the studies they refer to are not cited. Another example: "The subjects included in this report are a subset of those reported in an earlier study". Cite them accordingly not only in this statement but throughout the text.
Minor.
6. Please acknowledge that the literature search was conducted independently by two authors in the methods section.
7. The authors did not provide evidence/discussed why these studies are not biased because of different measurement techniques, according to them.
8. Figure 3 shows some strange behaviour. Identification, searching, included are not written in the right format.
9. Please correct “The effect size estimates are not available.” to “The effect size estimates were not detailed in the original publication”.
10. Please check that all abbreviations are defined the first time they are used in the text (eg. HIV). Usually, abbreviatures in Tables are also defined in the footnotes of the Table instead of the Table per se. even if they were previously defined in the main text.
11. I do not understand the point the authors try to make with the following statement. Please, check the English grammar: “All these studies have the advantage of being able to count on them and to know the relationship between pesticide exposure in pregnant women and its effect on the respiratory system of their children”.
Author Response
Thank you for your revision. The changes have been made. Please see the attachment.

This manuscript is a resubmission of an earlier submission. The following is a list of the peer review reports and author responses from that submission.
Round 1
Reviewer 1 Report
The manuscript written by María Isabel et al. reviewed recent published studies on the relationship between maternal pesticides exposure and the occurrence of respiratory disease and symptoms in their children.
There are some issues need to be solved and improved:
1. English writing needs to be improved throughout the whole text.
2. Section 2 needs to be well refined, somehow repeating, going back and forth.
3. Careful read and correction of English expression and spelling are needed: Line 21, ‘potencial’; Line 26, ‘ty’; Line 135, COPD is not short for chronic lung disease, Line 122 ‘perinatal, and perinata exposure’ seems to be repeating. And so on.
4. Introduction was not well introducing the diseases going to be discussed in the results, e.g. cleft palate.
5. Studies introduced in Section 3.1 are not well related to the aim of this review, but more about childhood exposure related effects.
6. Effects of causing infertility and low body weight described in paragraphs of Line 141-149 were not matching the title of the section 3.2 or even the title of the manuscript.
7. The number of related studies is somehow few and bit old, additionally, as above-mentioned (5 and 6), studies not well related to the aim were also included, making the evidence weaker.
Author Response
Thank you very much for your comments.
- English has been improved and revised by an expert.
- Section 2 has been rewritten.
- Sorry for the english mistakes, they have been corrected.
- Introduction has been improved and we have introduced the diseases that are discussed in the results, including cleft palate.
- The aim has been modified according to the articles included in the results and discussion.
- The title of section 3.2 and the general title has been modified according to the information that it contains.
- There is not so much information about this topic, for this reason some of the articles are a bit old. The aim of the study has been modified.

Reviewer 2 Report
Dear Authors the MS explored the impact of pesticides on potential adverse effects in pregnant women an potential complications in their children.
The topic is very actual and interesting. The MS is well written with interesting conclusions showing Chronic exposure 16 to pesticides in fetuses is associates with chronic respiratory symptoms and disease.
Methods are rigorous and in line with PRISMA guidelines. Tables and figures respect the guideline for a systematic review.
I have only minor suggestion. At page 1 line 32 authors should introduce recent evidence on tossicity and impact in pregnancy and fertility given by drugs as chemotherapy and oral supplementation used in pregnancy.
In this view it is appropriate to introduce the following references:
- Effect of Neuroendocrine Neoplasm Treatment on Human Reproductive Health and Sexual Function. J Clin Med 2022
Virginia Zamponi et al.
PMID 35887747
- Selenium Supplementation in Pregnant Women with Autoimmune Thyroiditis: A Practical Approach. Nutrients. 2022 May 27;14(11):2234. doi: 10.3390/nu14112234.
Minnetti M et al.
PMID: 35684035
- Selenium and reproductive function. A systematic review.
Mirone M,
J Endocrinol Invest. 2013 Nov;36(10 Suppl):28-36.
PMID: 24419057
- Selenium supplementation in the management of thyroid autoimmunity during pregnancy: results of the "SERENA study", a randomized, double-blind, placebo-controlled trial.
Mantovani G,
Endocrine. 2019 Dec;66(3):542-550. doi: 10.1007/s12020-019-01958-1. Epub 2019 May 25.
PMID: 31129812
Author Response
Thank you very much for your comments. The references have been added.

Reviewer 3 Report
The health risks associated with pesticides exposure during pregnancy is an important topic. There is an extensive academic literature on it, that require careful interpretation. Authors carried out a narrative review on this field.
This work provides nothing new because a narrative review is far less systematic and rigorous than existing studies (systematic reviews, metanalyses).
The research question is not precise enough ("to determine the relationship between..."), search & selection criteria are not detailled.
The type of review should be announced in title and abstract. The title did not reflect the contents of the article.
Author Response
Thank you very much for your comments.
The research question has been changed to: The aim of this study was to determine the relationship between pesticide exposure in pregnant women and its effect on the health of their children.
Selection process and analysis of results
The article selection process was carried out in three phases. In the first phase, articles were selected on the basis of their title and abstract, to subsequently check the type of study design and examine compliance with the selection criteria previously described. They were considered potentially relevant to the topic when the conclusions and results contained information on the effects of endocrine disruptors on pregnant women and the consequences on the respiratory tract of their children. The full-text papers were then reviewed and the methodological quality of the studies was assessed.
The most significant data from all studies were collected and grouped by category using a manual data extraction sheet. The analysis variables extracted were: (i) author/s and year of publication, (ii) aim, (iii) methodology, (iv) most relevant results.
The title states that this is a literature review.

Round 2
Reviewer 1 Report
After the first round revision, the manuscript has been improved. There are still some points could be improved:
1. The logic in the abstract is somehow weird, please develop it.
2. Small mistakes: Space before references 14, 18, 19, 26, 27 and 2.2.2; uniform if you want to use period in the subtitles or not.
3. In my opinion, Exclusion criteria is not necessary, since it is just the opposite to the Inclusion criteria, repeating. This information (inclusion criteria) is repeated again in the first paragraph of 2.3 except for the databases information. To appear once should be enough under the same section of 2.
4. There is a big possibility of better organizing the Discussion.
Reviewer 3 Report
I remain of the same view: a narrative review on this topic presents little interest.
Authors have changed the title but their new proposal cannot be accepted because it il not enough precise (NARRATIVE REVIEW / RESPIRATORY EFFECTS IN CHILD).